# Problematic Internet Use and Psychological Distress in High School Students

**DOI:** 10.3390/healthcare13243231

**Published:** 2025-12-10

**Authors:** Irati Becerril-Atxikallende, Joana Jaureguizar, Nuria Galende

**Affiliations:** Department of Developmental and Educational Psychology, University of the Basque Country (EHU), 48940 Leioa, Spain

**Keywords:** adolescence, addiction, anxiety, depression, ICT, psychological distress

## Abstract

**Background/Objectives**: The frequent and uncontrolled use of digital devices has resulted in phenomena such as technological addiction and problematic ICT use, especially after the pandemic. This has been associated with several factors related to psychological distress in young adults, but less is known about the subject in adolescents. Thus, the aim of this study is to analyze the relationship between problematic Internet use and psychological distress factors in high school students and examine whether these variables differed when gender and academic grade level were considered. **Methods**: A quantitative, cross-sectional, and descriptive–correlational study was employed. A total of 2048 students from the Basque Country, aged between 11 and 17 years old, completed an online self-report questionnaire composed of demographics and ad hoc items, the Problematic Internet Use Scale (PIUS), and selected subscales from the Child and Adolescent Assessment System (anxiety, social anxiety, and depression). Descriptive statistics, Pearson correlation analyses, independent-sample ANOVA, Bonferroni post hoc tests, and independent-sample t tests were conducted. **Results**: Even though no differences were found between males and females when PIUS was analyzed, significant differences were found between students from different academic grade levels, whereby those from higher levels presented higher rates of problematic Internet use. Significant correlations were found between PIUS and depression, anxiety, and social anxiety. Furthermore, those who showed more problematic Internet use also presented higher anxiety, social anxiety, and depression levels. **Conclusions**: Adolescents in higher grade levels tend to exhibit a higher incidence of problematic Internet use. Consequently, intensive and uncontrolled Internet usage has been linked to poorer mental health. The findings underline the importance of promoting digital literacy among adolescents. These results highlight the importance of approaching psychological distress through prevention initiatives and emphasize the protective role that both schools and families play in promoting healthier and more balanced Internet use among adolescents.

## 1. Introduction

Over the past decade, studies on the use of digital technologies among adolescents and children have grown significantly [1,2], especially after the COVID-19 pandemic [3,4,5]. The use of Information and Communication Technologies (ICTs) has become a widespread phenomenon among children and adolescents, turning into an essential aspect in their social [6] and educational spheres [7]. According to the recently published report issued by the Organization for Economic Co-operation and Development (OECD), 96% of 15-year-olds in OECD countries have access to a laptop, a desktop computer, or a tablet at home, and 98% have a smartphone with Internet access [8]. These data coincide with those published in the UNICEF report [9] on the digitization of young people aged between 11 and 18 in the Basque Country, where 96.80% have a mobile phone with Internet access, 91.30% connect every day or almost every day, and 31% use the Internet for more than 5 hours a day on weekdays, with this percentage increasing to 46.4% on weekends.

While the daily presence of ICTs in today’s society is undeniable, the way in which they are used, especially among children and young people, has raised serious concerns in the fields of education and health, with a growing number of studies showing the risks that dysfunctional use can present to human development [9,10,11]. Frequent, prolonged, and uncontrolled use of digital technologies can lead to compulsive, addictive, or problematic use, interfering with people’s daily lives, especially among young people [10,12]. In this context, both the frequency and the time spent using digital technologies have been found to be relevant factors, as an increase in both is associated with a higher probability of problematic ICT use [13]. Recent studies have shown that these patterns of use are associated with decreased academic performance, deterioration in social relationships such as family relationships, reduced emotional regulation skills, and poorer mental health [11,14,15].

Mental health in childhood and adolescence has become a subject of great scientific interest, given that this is a vulnerable period in which minors build their identity and self-esteem, expand their network of interpersonal relationships, seek new sensations, behave riskily, have low levels of self-control, etc. In other words, they undergo multiple physical, psychological, emotional, and social changes that have a decisive impact on their adult lives [12,15,16,17]. International studies report an increase in mental health problems—such as difficulties in emotional regulation, sleep disorders, higher levels of loneliness, anxiety, depression, or stress, and low life satisfaction—especially after the COVID-19 pandemic, placing mental health as an emerging social challenge of great relevance [18,19]. Furthermore, the scientific literature has highlighted a consistent association between problematic ICT use and various indicators of psychological distress (including anxiety, social anxiety, and depression) in adolescence, as compulsive or non-regulated use of digital devices is associated with higher levels of depressive symptoms, anxiety, and loneliness, as well as a decrease in subjective well-being, less self-esteem, and poor sleep quality [4,20,21]. These associations can be conceptualized through models such as I-PACE and compensatory use theories, which suggest that young people may use digital media or ICTs to avoid, manage, or relieve negative emotions, real-life problems, or stressful situations. Over time, this pattern may progressively lead to a less controlled use of ICTs and the emergence of addictive behaviors, which have been consistently associated with higher levels of psychological distress symptoms [22,23]. Adolescents experiencing mental health challenges—such as depression and anxiety—are more likely to engage in Problematic Internet Use (PIU) [20,24]. In this sense, PIU may function as a coping strategy, offering temporary relief from emotional difficulties and everyday stressors by providing an escape into the digital environment [25] and avoid real-life stress or uncomfortable social interactions [26]. Collectively, these findings suggest that pre-existing psychological vulnerabilities significantly contribute to the development of PIU.

On the other hand, in order to identify the profile of minors who make problematic use of digital media or those who are most vulnerable to developing this type of behavior, sociodemographic variables such as gender and age have been considered in multiple studies. However, the findings are not consistent. While some studies claim that males have higher levels of problematic Internet use and a greater risk of developing technological addiction than females [27,28], other studies indicate that females tend to report higher levels of problematic digital media use compared to males [29,30], and still others show that there are no differences between males and females [31,32]. Likewise, age is another variable that has been analyzed in different studies and has shown discrepancies. Some authors indicate that the older adolescents are, the higher the level of problematic ICT use [29]. These conclusions coincide with the results of other authors, showing that it is during middle and late adolescence that the highest risk rates are found compared to adults [33]. However, other authors have found no differences between adolescents of different ages or stages [30].

In this context, it is essential to further investigate the use of digital technologies among children and adolescents, and the relationship between the problematic use of digital technologies, sociodemographic variables, and variables related to psychological distress in this population [34]. Therefore, this study aims to examine patterns of Internet use in a sample of high school students, distinguishing between non-problematic and problematic use. It also seeks to analyze whether there are differences in Internet use based on gender and academic grade level, as well as to explore the relationship between such use and internalizing variables linked to psychological distress. Finally, it aims to explore and compare levels of anxiety, social anxiety, and depression between students who use the Internet moderately and those who use it problematically.

## 2. Methods

### 2.1. Study Design and Participants

This study used a cross-sectional and descriptive-correlational design. The sample comprised 2048 high school students: 12.40% (n = 253) were from public or state schools and 87.60% (n = 1795) from state-subsidized private schools or charter schools of the Basque Country (Spain). Regarding gender, 48.50% (n = 993) of the students were males, 50.40% (n = 1033) were females, and 1.10% (n = 22) were non-binary. Participants were between 11 and 17 years old (M = 13.61, SD = 1.20). Regarding academic grade level, 27.10% (n = 556) were in the first grade, 25.30% (n = 519) were in the second grade, 25.70% (n = 526) were in the third grade, and 21.80% (n = 446) were in the fourth grade.

### 2.2. Measures

For data collection, sociodemographic information from participants was collected, including variables such as gender, age, and academic grade level. Additionally, ad hoc questions were designed to obtain information about the digital devices participants owned and the age at which they first received them.

The Internet Problematic Use Scale (PIUS) [35] was used for assessing adolescents’ Internet use through 11 items. Items were formulated in the first person and rated on a Likert scale from 0 to 4, with 0 indicating *Totally disagree* and 4 indicating *Totally agree*. The total score range was between 0 and 44. Higher scores meant greater Internet problematic use. The internal consistency of the scale in our sample was α = 0.82. As the PIUS does not have an established cut-off point, and due to the absence of a consensus criterion, the procedure described by the original authors was replicated, estimating the optimal cut-off point using an ROC curve analysis. Participants were classified into two groups (moderate use and problematic use) based on specific behavioral criteria, and then the sensitivity and specificity values were calculated for different cut-off points. In this study, the analysis showed an area under the curve (AUC) of 1.000, indicating perfect discrimination between the compared groups. The cut-off point corresponding to 15.50 had a sensitivity of 1.000 and a specificity of 1.000 (1 − specificity = 0.000). Based on these values, the Youden index was calculated using the formula J = sensitivity + specificity – 1, obtaining a value of J = 1.000. The adjacent cut-off points (14.50 and 16.50) showed lower Youden values (0.944 and 0.896, respectively), confirming the optimal cut-off point of 15.50. However, as this is a scale with whole scores, a value of 16 was adopted as the original authors proposed [35] and based on the distribution of scores observed in our sample.

The Children and Adolescents Assessment System (SENA) [36] questionnaire’s three subscales were used to assess psychological distress in minors. The selected subscales were *anxiety* (10 items), *social anxiety* (8 items), and *depression* (14 items). This instrument consists of self-report items where participants must indicate the frequency with which they have experienced each of the behaviors described in the items. The response option is a Likert scale with five options, from 1 to 5, where 1 indicates *Never or almost never* and 5 indicates *Always or almost always*. Cronbach’s alphas in the present sample were α = 0.90 for *anxiety*, α = 0.84 for *social anxiety*, and α = 0.92 for *depression*.

### 2.3. Study Procedure

After the study was approved by the ethics committee of the University of the Basque Country (M10_2023_297), schools were randomly selected from the complete list of schools in the Basque Country. The researchers provided the school administrations with a written consent letter explaining the study’s objectives, design, and procedure. After receiving their signed consent, participants and their families received a written informed consent with the study information. Only participants who obtained parental consent and provided their own assent participated voluntarily, with the knowledge that they could withdraw at any point without negative consequences. Participants were asked to complete an online self-report questionnaire in their classrooms, where a researcher was always present. Data were collected between September 2024 and April 2025.

### 2.4. Statistical Analysis

Data analysis was performed using SPSS version 29.0. (SPSS, Chicago, IL, USA ) Descriptive statistics were used to describe the sample. The Kolmogorov–Smirnov test, Levene’s test, and skewness and kurtosis values were considered to verify the suitability of the use of parametric tests. Statistical significance level was set at *p* < 0.05. Correlations between variables were conducted with Pearson correlation analyses. Independent-sample *t* test was used for comparisons between two groups, and one-way ANOVA for comparisons involving more than two groups. ANOVA’s analysis was followed by Bonferroni post hoc tests to identify differences between specific groups. Effect size was interpreted based on Cohen’s (1988) [37] guidelines: 0.20 to 0.50, small effect; 0.50 to 0.80, intermediate effect; and 0.8 or higher, large effect.

## 3. Results

### 3.1. Descriptive Statistics on ICT Acquisition and Use

Most adolescents (85%, n = 1741) reported owning a personal smartphone, with a mean age at first acquisition of 11.21 years (SD = 2.83). A total of 42.80% (n = 876) had their own tablets, and the mean age of obtaining their first tablet was 8.51 years (SD = 1.19). Apart from school laptops or computers, 37.90% (n = 777) of the minors said they had their own electronic device, and the mean age of first laptop or computer acquisition was 10.77 years (SD = 2.08). A total of 28% (n = 574) said they had their own smartwatch, and the mean age of obtaining their first smartwatch was 10.54 years (SD = 2.36).

In addition to analyzing the percentage of students who had each of the different types of devices and the average age of their first acquisition, the use of the Internet by minors was also observed: 70.40% (n = 1442) reported moderate use of the Internet, and 29.10% (n = 595) engaged in problematic use.

### 3.2. Problematic Internet Use by Gender and Academic Grade Level

Problematic Internet use was analyzed according to students’ gender and academic grade levels to determine whether there were significant differences between males and females, as well as among first- to fourth-grade students. The group of students who identified themselves as non-binary was excluded from the comparative analysis due to their small sample size. Likewise, the results showed that there were no differences between males and females [*t*(2013) = 0.583, *p* = 0.560, *Cohen’s d* = 0.026].

On the other hand, significant differences were found between students in different academic grade levels when analyzing problematic Internet use [*F*(3) = 13.051, *p* < 0.001, ƞ^2^ = 0.019]. Post hoc comparisons using Bonferroni adjustment revealed significant differences between first- and third-grade students (*p* < 0.001, *Cohen’s d* = −0.291), as well as between first- and fourth-grade students (*p* < 0.001, *Cohen’s d* = −0.313). In both cases, students from upper grades showed higher levels of problematic Internet use compared to first-grade students (Table 1).

Similar results were found when second-grade students were analyzed. Significant differences were found between second- and third-grade students (*p* = 0.002, *Cohen’s d* = −0.221, small), and between second- and fourth-grade students (*p* < 0.001, *Cohen’s d* = −0.244, small). Table 1 illustrates how both third- and fourth-grade students obtained higher scores in problematic Internet use, compared to second-grade students.

### 3.3. Problematic Internet Use and Internalizing Variables

Table 2 illustrates correlations between problematic Internet use and internalizing variables, including anxiety, social anxiety, and depression. Significant correlations were found between problematic Internet use and variables related to psychological distress. The highest correlations were found between problematic Internet use and depression, followed by anxiety and social anxiety.

In addition to the correlations, each of the variables related to psychological distress was analyzed in terms of the type of Internet use reported by students. Table 3 presents the descriptive statistics for internalizing symptoms—anxiety, social anxiety, and depression—stratified by participants’ level of problematic Internet use.

Independent-sample *t*-tests revealed that significant differences were found between students who use the Internet moderately and those who use it problematically when analyzing the anxiety variable [*t*(1723) = −10.08, *p* < 0.001, *Cohen’s d* = −0.536]. As shown in Table 3, students who reported problematic use of the Internet showed higher levels of anxiety, compared to those who used it moderately, and manifested lower anxiety levels.

When the social anxiety variable was analyzed, significant differences were also found between moderate and problematic Internet users [*t*(1759) = −8.85, *p* < 0.001, *Cohen’s d* = −0.466]. Specifically, adolescents who reported problematic Internet use reported higher levels of social anxiety compared to moderate users, showing lower social anxiety scores.

Significant differences were also found between students who reported moderate and problematic use when observing the depression variable [*t*(1569) = −11.82, *p* < 0.001, *Cohen’s d* = −0.709]. Minors who engaged in problematic Internet use were also the ones who obtained higher scores in the depression variable, in contrast to adolescents who reported moderate use, presenting lower depression scores.

## 4. Discussion

The present study sought to analyze the percentage of high school students who owned digital devices, explore the types of Internet use, and determine whether usage differed based on gender and academic grade level. This research also aimed to examine the association between Internet use and internalizing variables related to mental health, as well as identifying how levels of psychological distress varied according to the different patterns of Internet use reported by adolescents.

The results showed that smartphones are the most widely used devices among students, as a majority reported ownership and 11 was the mean age at which they acquired their first smartphone. The results align with those reported in other studies, which also identified smartphones as the most commonly owned device among young people, with the average age at which they acquired their first mobile phone being approximately 11 years old [9,38,39]. Following mobile phones with Internet access, the most popular devices among the students in the sample were, in descending order, tablets, computers, and finally, smartwatches. Similar findings were observed in another study, although ownership percentages were higher, with tablets being the second most popular device after smartphones, followed by laptops and televisions [40]. However, another research has shown that, after smartphones, laptops are the most common device among teenagers, followed by tablets and finally gaming devices [38]. The differences can be explained by the type of device used in schools, as some facilitate the use of tablets and others the use of computers.

Another relevant aspect to consider is the considerable variability across studies in the prevalence of problematic Internet use among young people, which may be due to cultural, socioeconomic, or temporal factors [41]. In this regard, a recent study has shown that the number of young people who used the Internet problematically before the COVID-19 pandemic was lower, while during the pandemic and—particularly—after it, an upward trend was observed [3]. Although some authors have shown that most adolescents (62.60%) use the Internet in a problematic way [38], other studies have shown that the percentage of young people who report problematic Internet use is lower. According to some authors, this percentage is around 43.69% [3], while others estimate it to be around 40% [32]. Other experts [42,43] have shown that the percentage is even lower, around 24% and 23%, respectively. In this context, the findings of the present study represent an intermediate value, as around 30% of minors reported problematic Internet use. Contrary to expectations that most participants would make worse use of the Internet, the opposite was observed, i.e., most adolescents reported moderate or regular use of the Internet. These findings reinforce the conclusions of other studies, which also observed that 33% [44,45] or 28.90% [46] of young people use the Internet in a problematic way. Although minors in general do not typically report maladaptive or dysfunctional use of this tool, this does not mean that they are not at risk of developing such behaviors; it is therefore essential to highlight the protective role of certain factors, such as the family environment, relationships, or parental control over adolescents [32,47,48]. This 30% percentage is not insignificant, however, because it indicates that many teenagers have difficulty controlling the amount of time they spend online, neglecting other activities or their responsibilities in order to be connected, or even feeling uncomfortable when they cannot connect.

The available evidence reveals a notable disparity in findings regarding the differences in problematic Internet use between genders and academic grade levels or age groups. While some studies have concluded that girls exhibit greater problematic Internet use [3,49], others have observed that boys are more likely to use the Internet in a maladaptive manner [27,50]. In contrast, the sample analyzed in this study showed no significant differences when considering gender, as both girls and boys exhibited similar values in problematic Internet use. These findings align with those of other studies, which have found no differences based on gender [27,28]. This disparity in results could be explained by factors identified in various studies, which argue that gender alone is not a consistent determinant of problematic use of digital technologies [51]. Currently, the term PIU does not discriminate against specific activities carried out by young people. Several studies have shown gender differences in specific internet behaviors [52]. Males tend to use digital media for entertainment purposes, such as playing video games, while females are more likely to use them to access social media platforms, socialize, and communicate with their friends [30,53]. From this point of view, our study suggests that PIU is similar in males and females, but, among other limitations of our study, the specific activities of each are not distinguished. Discrimination of screen-related activities should be considered when analyzing PIU, as pathological use may differ, as females with externalizing behavior are at risk of becoming overly involved in social media, and males with internalizing behavior are at risk of becoming withdrawn in video game-related disorders [54].

Besides gender, academic grade level was another variable of interest in the analysis of problematic Internet use. Previous literature has shown that problematic Internet use can vary depending on age, with some cases identifying a higher risk as age increases during adolescence [29]. According to some authors, older adolescents make greater problematic use of the Internet [3]. However, unlike the studies mentioned earlier [3,29], other studies have not found significant differences between minors of different ages or academic grade levels [30,50]. The findings of the present research are directly in line with those previous findings, as participants from the third and fourth grades, i.e., the older ones, scored higher on problematic Internet use compared to their younger peers, i.e., students from the first and second grades. One possible explanation for the results obtained is that upper-level high school students, in addition to using the Internet more frequently, are under less parental supervision compared to younger students [55]. These factors, combined with the fact that there is a strong association between increased time and frequency of digital media use and a higher probability of problematic use [13], could lead to older adolescents or those in higher academic grade levels being more vulnerable to developing problematic use of these tools. Considering that parental control is a fundamental factor in preventing risks associated with Internet use [56] and that such control tends to decrease as adolescents get older [55], it would be relevant and necessary for future research to assess the degree of supervision exercised by gathering such information from both the adolescents themselves and their parents.

Another research focus has been on the association between problematic Internet use and internalizing variables. In the present study, the strongest positive correlations were found between problematic Internet use and depression, followed by anxiety and social anxiety. Additionally, students who indicated higher scores in the Problematic Internet Use Scale also showed higher depression, anxiety, and social anxiety levels. The most remarkable difference between moderate and problematic Internet users was seen in depression, which can be considered an indicator of psychological distress, thus placing it as the most relevant factor in differentiating between the two user profiles. Similarly, other research has reached similar conclusions. According to some experts, youths who present severe anxiety or depressive symptoms also show greater problematic Internet use [43]. Similar results were found in other studies conducted with adolescents, where those with more problematic Internet use had higher scores on depression, suicide behaviors [57], and social phobia [45]. Moreover, a meta-analysis conducted with pre-COVID studies, as well as more recent research, has shown that problematic use of digital media is also associated with higher stress levels, as excessive or uncontrolled online activity may intensify emotional reactivity and difficulties in stress regulation [58,59]. On top of that, it is necessary to distinguish between moderate or regular use and problematic or dysfunctional use of digital technologies, as while problematic users experience higher levels of distress, regular users do not show those symptoms. Some authors have shown that associations between general use of digital technologies and mental health are small, inconsistent, and do not reflect a progressive worsening over time [34]. The results of the current study, in conjunction with those of previous research [43,45,57], are particularly relevant, as they show how more problematic Internet use is associated with greater psychological distress in different dimensions, i.e., they highlight the relationship between the use of digital technologies and the well-being of minors. A possible explanation proposed in the literature that has been associated with problematic use of digital media and psychological distress symptoms is that some young people use these tools as a strategy for emotional regulation and a means to seek distraction, find short-term relief, avoid real-life situations or compensate for a lack of face-to-face social interaction, potentially contributing to more problematic patterns of use [12,22,23,58,59]. All of this underscores the need to investigate the issue more deeply, reduce excessive Internet use among highly vulnerable populations, such as children and adolescents, and promote strategies to encourage moderate, safe, and critical use of the Internet as a preventive measure to mitigate its risks among them.

Despite the methodological rigor applied in this study, it is important to acknowledge some limitations. First, the cross-sectional design provides only a general overview of the reality analyzed, without showing the evolution of problematic Internet use among adolescents over time. Future research with a longitudinal approach could offer a deeper understanding of the evolution of these patterns and their implications for mental health. Second, the study employed a quantitative approach, which enables the identification of associations between variables but does not delve into the participants’ subjective experiences and perceptions in depth. The incorporation of qualitative techniques in future research would contribute to a richer and more contextualized understanding of the meanings that adolescents attach to their relationship with problematic Internet use. The exclusive use of self-reports may be another limitation because of the possibility of shared method variance. Moreover, the absence of objective digital usage metrics or reports by external informants, such as parents or teachers, may limit the precision of behavioral assessment. Thus, including objective measures and involving other agents outside adolescents themselves should improve future studies. Although highly relevant variables such as anxiety, social anxiety, and depression were included, other factors should also be analyzed, such as self-esteem, emotional regulation, or social support, as there might be associations with problematic Internet use. Additionally, the reduced representation of students from public schools may limit the diversity of the sample and, consequently, the generalization of the results. Future research should include a larger sample of public school students and conduct a comparative analysis between public and state-subsidized private school students, as contextual factors such as socioeconomic level may play an important role. Finally, the possibility of bidirectional associations between problematic Internet use and psychological distress should be considered, as the study design does not allow for causal inferences. Considering these dimensions in future research will enable us to develop a more comprehensive and explanatory framework.

## 5. Conclusions

The results showed that most adolescents own their own smartphones, and to a lesser extent, tablets, computers, and smartwatches. Furthermore, most participants reported moderate Internet use, and although no significant differences were found between males and females, when analyzing problematic Internet use, higher levels of dysfunctional use were identified among students in higher grades (third and fourth grades). On the other hand, associations were found between problematic Internet use and internalizing variables, with the strongest correlation observed with the depression variable, followed by anxiety and social anxiety. Moreover, it was observed that students who reported more problematic Internet use also showed higher rates of psychological distress. These findings contribute to the existing literature by pointing out that, among the adolescent population, their academic level is a more significant factor than gender in explaining problematic Internet use, confirming the close relationship between such use and mental health indicators. Given this context, the results highlight the need for further research into minors’ use of digital tools and the role these play in their personal, social, and academic contexts. Likewise, they emphasize the need to design and implement prevention and intervention strategies targeting minors, their families, and teachers that promote healthy digital habits, foster digital literacy, and enhance the psychological well-being of minors.

## Figures and Tables

**Table 1 healthcare-13-03231-t001:** Descriptive statistics of problematic Internet use by gender and academic grade level.

	Total		Gender				Academic Grade Level							
			Males		Females		First		Second		Third		Fourth	
	M	SD	M	SD	M	SD	M	SD	M	SD	M	SD	M	SD
Problematic Internet Use	12.41	7.18	12.47	7.28	12.27	7.01	11.25	7.56	11.63	6.86	13.34	7.06	13.44	6.74

**Table 2 healthcare-13-03231-t002:** Correlations between problematic Internet use and internalizing variables.

	1	2	3	4
1. Internet Problematic Use	-			
2. Anxiety	0.305 **	-		
3. Social anxiety	0.257 **	0.618 **	-	
4. Depression	0.410 **	0.753 **	0.515 **	-

** *p* < 0.001.

**Table 3 healthcare-13-03231-t003:** Descriptive statistics of internalizing variables by level of problematic Internet use.

	Total		Problematic Internet Use			
			Moderate Use		Problematic Use	
	M	SD	M	SD	M	SD
Anxiety	2.80	0.88	2.67	0.86	3.14	0.88
Social anxiety	2.55	0.82	2.43	0.80	2.83	0.82
Depression	2.09	0.76	1.94	0.68	2.46	0.82

## Data Availability

Dataset available on request from the authors.

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
