# Peer review of "Problematic Internet Use and Psychological Distress in High School Students"

_healthcare, 2025, doi:10.3390/healthcare13243231_

Round 1

Reviewer 1 Report

Comments and Suggestions for Authors

The article addresses the important issue of problematic Internet use among adolescents.

The abstract contains essential information, but it would be beneficial to add the age of the respondents to the description of the study group, as this is important because the analyzed correlations proved to be age-related. Although it is mentioned that high school students were surveyed, the age of high school students may be varied in different countries.

The introduction provides a very good explanation of the phenomenon of problematic use of digital devices and technologies, which has become particularly acute since the COVID pandemic, and the significance of this phenomenon in the development of adolescents. However, the role of distress is not sufficiently explained here, although the role of anxiety and depression is indicated. The introduction should be completed with more detailed data on distress in order to maintain consistency and coherence with the title of the article (or add information that distress, as understood here, includes anxiety, social anxiety, and depression).

Previous studies have not clearly demonstrated a link between sociodemographic factors such as gender and age and problematic Internet use. Given the risks associated with this phenomenon and the ambiguity of the data available to date, the analyses undertaken are reasonable.

The description of the study group states that “12.40% (n = 253) were from public or state schools and 87.60% (n = 1795) from state-subsidized private schools or charter schools in the Basque Country (Spain)”. What was the basis for this sample selection and is it representative?

The research tools were described in a complete and transparent manner.

The research procedure lacked information on whether the research was conducted using the paper-and-pencil method (it can be assumed that this was the case, but it was not explicitly stated), as well as what the criteria for inclusion and exclusion from the research were – was consent to participate in the research sufficient?

The statistical analyses were correctly selected and the results were presented in a clear and transparent manner. The discussion was generally conducted correctly, although the analysis of the relationship between problematic Internet use and stress needs to be deepened. The text indicates the limitations of the study. The conclusions are consistent with the results obtained and the subject of the article, but they are too extensive – it is worth considering shortening this part of the text (e.g., omitting the first two sentences) so that it is not a summary, but precisely conclusions.

The literature has been well selected.

The text is interesting and cognitively valuable. I recommend its publication after the corrections indicated in the review have been made.

Reviewer 2 Report

Comments and Suggestions for Authors

Major concerns

1.  the Theoretical Framework and Strengthen the Study’s Contribution

The introduction offers a broad overview of adolescents’ digital technology use, but it would benefit from a clearer conceptual grounding. At present, the narrative focuses on prevalence and descriptive patterns without sufficiently drawing on existing theoretical models that explain why problematic use emerges or how it relates to mental health.

It would enhance the manuscript to integrate well-established frameworks such as the I-PACE model, compensatory/escape-based models of Internet use, or developmental theories related to emotional regulation. These perspectives could help the reader understand the mechanisms linking PIU with depression, anxiety, and social anxiety.

Additionally, the study’s distinctive contribution could be stated more explicitly. Potential strengths worth highlighting include the ROC-based determination of a PIUS cut-off score, your large regional sample, and the focus on grade level rather than chronological age, which is less commonly examined in the literature.

2. Provide Additional Detail and Transparency in Methodology

The methodology is generally sound, but a few areas would benefit from fuller explanation to enhance transparency.

  • ROC Analysis: Readers need more information about the diagnostic performance of the cut-off score you established. Reporting the AUC value, sensitivity, specificity, and the Youden index would enable a clearer understanding of the scale’s discriminative ability in your sample.

  • Sampling Structure: Because the majority of participants are from subsidized private schools, it would be helpful to acknowledge and discuss how this may influence the generalizability of the findings, particularly given links between socioeconomic status, parental monitoring, and digital behavior.

  • Multiple Statistical Comparisons: Given the number of t-tests and correlations performed, it would be advisable to address the potential for Type I error. If corrections for multiple testing were not applied, briefly explain the rationale for this decision.

  • Self-Report Measures: Since all information was self-reported, including both PIU and mental health indicators, the possibility of shared method variance should be acknowledged as a limitation.

These clarifications will strengthen the methodological rigor and make the findings more robust.

3. Expand Interpretation of Findings in the Discussion

Your results are clearly presented, but the discussion would benefit from a deeper interpretation of the patterns observed, particularly regarding why older adolescents and those experiencing psychological distress may be more vulnerable to problematic use.

Existing research suggests that developmental transitions during mid-to-late adolescence (increasing autonomy, greater social pressures, identity experimentation) and reduced parental monitoring may contribute to higher PIU risk. Additionally, young people with anxiety or depression may engage in online activities for avoidance, emotional regulation, or social substitution. Integrating these perspectives will enrich your interpretation and help connect your findings to broader discussions in the field.

The null findings regarding gender also deserve more reflection. Many studies show that gender differences depend on the type of online activity (e.g., gaming vs. social networking). A brief discussion of this nuance would help contextualize your results and clarify why gender may not be a consistent predictor of PIU in all populations.

4. Strengthen the Limitations and Future Directions

Your limitations section is relevant but could be expanded. In addition to the cross-sectional design and exclusive reliance on quantitative measures, consider noting:

  • Limited representation of public schools, which may affect socioeconomic diversity.

  • Absence of objective digital usage metrics or parental/teacher reports.

  • The likelihood of bidirectional associations between PIU and psychological distress.

These additions will provide a more balanced view of the study’s scope and help guide future research efforts.

Minor Concerns

1. Language and Presentation

Some expressions are non-idiomatic (e.g., “punctuations” instead of “scores,” “made problematic use” instead of “engaged in problematic use”), and certain paragraphs would benefit from shorter sentences to improve readability.

2. Tables and Possible Figures

The tables are clear and informative. It would be beneficial to include confidence intervals for key estimates. Although optional, incorporating one or two figures—such as a correlation heatmap or grade-level comparisons—could make the results more visually accessible.

3. Abstract and Conclusion

The abstract could more clearly emphasize the key contribution of the study and the practical implications of your findings. Similarly, the conclusion could briefly reiterate the importance of the strongest associations observed (particularly with depression) and how these insights might inform school-based or family-oriented prevention programs.

Assessment

Your manuscript addresses an important issue with clear public health implications. The dataset is strong, the analysis is generally appropriate, and the topic is of considerable interest to researchers, practitioners, and policymakers. With improvements in theoretical framing, methodological detail, and interpretative depth, this work will be well positioned for publication.

Comments on the Quality of English Language

The manuscript is generally well written and clear, with an appropriate academic tone throughout. The meaning is easy to follow, and the overall structure supports comprehension. However, several areas would benefit from moderate language polishing to improve fluency and precision. A few expressions are non-idiomatic, some sentences are overly long, and certain terms (e.g., “punctuations,” “made problematic use”) could be replaced with more standard academic phrasing. There is also occasional repetition in the introduction and discussion that could be streamlined for clarity. These issues do not impede understanding, but revising them would enhance readability and the overall quality of presentation. Overall, the English is of good quality and requires only a moderate level of editing.

Round 2

Reviewer 2 Report

Comments and Suggestions for Authors

Thank you for the notes and the revision. The revised manuscript is now scientifically sound and addressing a highly significant to the public health issue. To ensure seamless publication, recheck your manuscript - revisions on these key areas.

- Conceptual design, formally integrate theoretical models to explain the mechanisms linking PIU and psychological distress

- Methodological design, enhance transparency by including the full ROC metrics and formally acknowledging shared method variance and the private school majority as limitations

- Interpretive debate, enrich the discussion by linking PIU in older adolescents to reduced parental supervision and contextualizing the null gender finding by discussing differences in online activity types

Addressing these points will significantly elevate your manuscript's impact and clarity.

Comments on the Quality of English Language

The manuscript is generally well written and clear, with an appropriate academic tone throughout. The meaning is easy to follow, and the overall structure supports comprehension. However, several areas would benefit from moderate language polishing to improve fluency and precision. Overall, the English is of good quality and requires a bit of checking and editing.
